# Risk Prediction of RNA Off-Targets of CRISPR Base Editors in Tissue-Specific Transcriptomes Using Language Models

**DOI:** 10.3390/ijms26041723

**Published:** 2025-02-18

**Authors:** Kazuki Nakamae, Takayuki Suzuki, Sora Yonezawa, Kentaro Yamamoto, Taro Kakuzaki, Hiromasa Ono, Yuki Naito, Hidemasa Bono

**Affiliations:** 1Genome Editing Innovation Center, Hiroshima University, Higashi-Hiroshima 739-0046, Japan; onohrms@hiroshima-u.ac.jp; 2Graduate School of Integrated Sciences for Life, Hiroshima University, Higashi-Hiroshima 739-0046, Japan; d233528@hiroshima-u.ac.jp (T.S.); d246887@hiroshima-u.ac.jp (S.Y.); 3Genome Analytics Japan Inc., Tokyo 162-0067, Japan; yamaken37.the.answer@gmail.com; 4Independent Researcher, Tokyo 162-0831, Japan; taro.kakuzaki@gmail.com; 5Database Center for Life Science, Joint Support-Center for Data Science Research, Research Organization of Information and Systems, 178-4-4 Wakashiba, Kashiwa 277-0871, Japan; y-naito@dbcls.rois.ac.jp

**Keywords:** CRISPR, base editing, RNA off-target effects, machine learning, tissue-specific transcriptome, PiCTURE, PROTECTiO

## Abstract

Base-editing technologies, particularly cytosine base editors (CBEs), allow precise gene modification without introducing double-strand breaks; however, unintended RNA off-target effects remain a critical concern and are under studied. To address this gap, we developed the Pipeline for CRISPR-induced Transcriptome-wide Unintended RNA Editing (PiCTURE), a standardized computational pipeline for detecting and quantifying transcriptome-wide CBE-induced RNA off-target events. PiCTURE identifies both canonical ACW (W = A or T/U) motif-dependent and non-canonical RNA off-targets, revealing a broader WCW motif that underlies many unanticipated edits. Additionally, we developed two machine learning models based on the DNABERT-2 language model, termed STL and SNL, which outperformed motif-only approaches in terms of accuracy, precision, recall, and F1 score. To demonstrate the practical application of our predictive model for CBE-induced RNA off-target risk, we integrated PiCTURE outputs with the Predicting RNA Off-target compared with Tissue-specific Expression for Caring for Tissue and Organ (PROTECTiO) pipeline and estimated RNA off-target risk for each transcript showing tissue-specific expression. The analysis revealed differences among tissues: while the brain and ovaries exhibited relatively low off-target burden, the colon and lungs displayed relatively high risks. Our study provides a comprehensive framework for RNA off-target profiling, emphasizing the importance of advanced machine learning-based classifiers in CBE safety evaluations and offering valuable insights to inform the development of safer genome-editing therapies.

## 1. Introduction

Genome-editing technologies have revolutionized biological research and therapeutic strategies by enabling precise genetic modifications in various organisms [1,2,3,4,5,6,7,8,9,10]. CRISPR-based base editors (BEs) are particularly noteworthy for their ability to introduce site-specific nucleotide substitutions without causing double-strand breaks [8,9,11], which is particularly advantageous for introducing efficient substitutions in clinical ex vivo and in vivo applications [12,13,14]. Among BE tools, cytosine BEs (CBEs) have received attention for their capacity to convert cytosine (C) to uracil (U), which has the same base-pairing characteristics as thymine (T) in dsDNA [9,15], facilitating targeted gene knockout via the introduction of a premature stop codon and the correction of pathogenic single-nucleotide polymorphisms (SNPs) [9,16]. However, several studies have reported that CBEs generate unintended C-to-U conversions in RNA, termed RNA off-target effects, which pose a potential safety concern [17,18,19,20,21]. Such RNA off-targets may elicit undesirable cellular responses, thereby compromising the safety and efficacy of CBE-based interventions.

While various strategies to mitigate RNA off-target effects have been developed [18,19,20,21], relatively little attention has been paid to the comprehensive, qualitative characterization of RNA off-target events. For example, canonical RNA off-targets often occur at known motifs, such as the ACW motif (where W represents A or T/U) [19,20]; however, non-canonical RNA off-targets that do not follow the ACW pattern remain poorly understood, and the field lacks a unified framework to systematically assess them. A thorough understanding of both canonical and non-canonical off-target events is critical for robust risk assessment and broader CBE applications.

To address this research gap, we developed the Pipeline for CRISPR-induced Transcriptome-wide Unintended RNA Editing (PiCTURE), a standardized and quantitative platform designed to evaluate RNA off-target effects, including non-canonical events. To refine predictive capabilities for CBE-based RNA off-targets, we leveraged recent advances in machine learning, notably the emergence of language models such as DNABERT-2 [22]. By integrating the output of PiCTURE with a fine-tuned DNABERT-2 model, we established a robust framework for RNA off-target risk assessment. In addition, we introduce Predicting RNA Off-target compared with Tissue-specific Expression for Caring for Tissue and Organ (PROTECTiO), a framework that estimates tissue-wide off-target risks by analyzing transcriptomic data with tissue-specific expression profiles. Collectively, these approaches offer a standardized methodology for assessing RNA off-target risks, marking a significant step toward more reliable CBE-related safety evaluations and facilitating further advancements in base-editing specificity and therapeutic applicability.

## 2. Results and Discussion

### 2.1. A Uniform Pipeline for Detecting RNA Off-Targets

We first employed PiCTURE (Figure 1) to detect RNA off-target events in CBE-treated samples. We analyzed 13 RNA-sequencing (RNA-seq) datasets, including samples treated with nCas9, BE3, and BE4-rAPOBEC1, as reported previously [20,21]. Consistent with previous findings [20], our analysis of BE3-treated samples revealed numerous C-to-U(T) substitutions (corresponding to G-to-A on the complementary strand) (Appendix A). By evaluating the 101-bp sequences flanking these substituted cytosines, we identified sequence motifs that influence substrate specificity (Appendix A). Notably, the 5′ adjacent nucleotide strongly influenced the likelihood of C-to-U(T) conversions, with cytosines flanked by A or T (W) at the 5′ side being more frequently edited (Appendix A). Additionally, the 3′ adjacent nucleotide showed a weak bias toward A or T (W). Although a previous report highlighted the ACW motif as being associated with RNA off-targets [20], our findings suggest that a broader WCW motif may better capture the substrate preference for unintended RNA editing in human cells. The WCW motif has been recently reported in RNA off-targets in BE3 and YE1-BE3-FNLS transgenic mice [17], supporting our finding that CBE-based RNA off-targets preferentially lie within the WCW RNA motif.

To obtain further support for this observation, we performed the same analysis on samples treated with BE4-rAPOBEC1 or nCas9 (Appendix A). We considered samples treated with Cas9 as negative controls because the substitutions detected in the nCas9 samples were expected to result from spontaneous mutations in the transcriptome. Similarly to BE3, BE4-rAPOBEC1-treated samples displayed motif-dependent C-to-U(T) substitutions, whereas no such motif enrichment was observed in the nCas9-treated samples. A comparison of the frequency of A or T nucleotides adjacent to converted Cs between BE4-rAPOBEC1 and nCas9 samples revealed a significantly higher frequency in the BE4-rAPOBEC1 samples than in the nCas9 samples (Appendix A). From these findings, it follows that, as noted above, not only the ACW motif but also related motifs such as TCW, WCA, and WCT—collectively represented by the broader WCW motif—can be interpreted as characteristic sequence contexts where BE3- and BE4-induced RNA off-target events occur frequently. A chromosome-wide analysis of C-to-U(T) conversions confirmed that BE4-rAPOBEC1-induced events were widely distributed (Appendix A) and significantly more abundant than those observed in nCas9-treated samples (Appendix A). These results demonstrate that PiCTURE enables the quantitative detection of C-to-U(T) substitutions, including CBE-induced RNA off-target events.

Next, we extracted datasets enriched for putative RNA off-targets from the identified C-to-U(T) conversion sites (Figure 2A). We first generated a negative dataset by collecting 40-bp or 101-bp sequences flanking each C-to-U(T) substitution site found in nCas9-treated replicates. We then generated a positive dataset from CBE-treated replicates by removing any sequences that overlapped with the negative dataset. For instance, using RNA-seq data from BE4-rAPOBEC1-transfected samples (SRR11561324, SRR11561298, and SRR11561273) and nCas9-transfected samples (SRR11561289, SRR11561314, and SRR11561326), we constructed a positive dataset. Among the 232,254 sequences in this positive dataset, 119,359 (51.4%) sequences contained the ACW motif (Figure 2B). Hereafter, we refer to these ACW-containing events as “canonical RNA off-targets”, and those lacking the ACW motif as “non-canonical RNA off-targets”. As mentioned above, we suggest that WCW motifs may also represent a key motif of unintended RNA editing. Indeed, examination of the positive dataset for the presence of WCW motifs showed that 159,545 out of 232,254 sequences (68.7%) contained a WCW motif (Figure 2B). This finding suggests that classification based on both ACW and WCW motifs may be valuable for detecting a broader range of CBE RNA off-targets.

### 2.2. Predicting RNA Off-Targets Using Nucleic Acid Language Models

We next aimed to construct a predictive model that does not solely rely on known motifs to identify RNA off-target-prone sequences. Recent advances in transformer-based nucleic acid language models [22,23,24,25,26] have enabled sophisticated sequence analyses. We adopted DNABERT-2 [22] as our base model and fine-tuned it using the negative and positive datasets derived from nCas9 and BE4-rAPOBEC1 samples, respectively, to develop a machine learning-based classifier for RNA off-target risk assessment (Figure 2C). In this development, we focused on a model that specifically learns to identify ‘CBE-induced substitutions’ rather than all possible substitution sites. We used nCas9-treated RNA-seq samples as the negative dataset to capture background mutations that are independent of CBEs, ensuring that our classifier targets genuinely CBE-driven effects and avoids under- or overestimating CBE-associated risk. First, we developed a model (hereafter referred to as the STL model) using datasets containing 78,944 positive and 21,790 negative sequences for training, 76,593 positive and 24,228 negative sequences for evaluation, and 76,717 positive and 23,476 negative sequences for testing. In each dataset, the positive samples outnumbered the negative samples more than 3.1-fold, and some sequences were duplicated across datasets. Despite these imbalances and overlaps, the STL model achieved an accuracy above 0.8 and outperformed classifiers that rely solely on ACW, WCW, or nonspecific (NNN) motifs in terms of accuracy, precision, and F1 score (Table 1).

However, the WCW motif-based classifier achieved better recall. To improve performance, we constructed a new dataset balanced for positive and negative sequences and without overlaps among the training, evaluation, and test sets. The new datasets contained 47,866 positive and 47,866 negative sequences for training, 10,257 positive and 10,257 negative sequences for evaluation, and 10,257 positive and 10,257 negative sequences for testing. We adjusted hyperparameters for better performance. We also shortened the input sequence length from 101 bp to 40 bp because we set “model_max_length” to 10 for our fine-tuning and, according to the DNABERT-2 manual, the recommended input sequence length should be four times “model_max_length”. Under these conditions, we developed a second model, hereafter referred to as the SNL model, which outperformed the WCW motif-based classifier in all performance metrics (Table 2).

We then benchmarked the STL model, SNL model, and motif-based classifiers against canonical and non-canonical RNA off-targets. Using the SNL model’s hold-out data (which were not used in training) to create datasets with and without the ACW motif, we randomly sampled 10 subsets and evaluated each classifier’s performance (Figure 3). For non-canonical RNA off-targets, the WCW motif-based classifier significantly outperformed the ACW motif-based classifier in all metrics, confirming the superiority of the WCW motif for detecting these events. Both the STL and SNL models significantly exceeded the performance of the WCW classifier in all metrics, demonstrating that language model-based predictions are particularly effective for detecting non-canonical RNA off-targets. The SNL model showed the highest accuracy, recall, and F1 score. Interestingly, the STL and SNL models also significantly outperformed motif-based classifiers for canonical RNA off-targets. While motif-based classifiers often exhibited very low precision (<0.3 on average), the STL and SNL models both achieved an average precision above 0.5, with the STL model reaching 0.72 on average. As precision reflects the proportion of predicted positives that are true positives, this improvement suggests that relying on motifs alone may lead to high false positive rates. Although identifying motifs from RNA off-target data is valuable, using these motifs directly as predictors may inflate false positives. In contrast, the STL and SNL models produce fewer false positives and thus may prevent overestimating the RNA off-target risk associated with CBEs.

### 2.3. Assessment of Tissue-Specific CBE Risk Using Transcriptome Databases

In addition to motif-based classifiers, we successfully developed machine learning-based classifiers that operate independently of known motifs, enabling unconstrained RNA off-target risk prediction on any transcriptome sequence. To demonstrate this capability, we created a safety assessment pipeline, PROTECTiO (Figure 4), aimed at evaluating tissue- and organ-specific CBE RNA off-target risks. PROTECTiO employs a tissue-specific gene expression database, RefEx [27], derived from human RNA-seq data. For each gene with elevated expression in a specific tissue, we obtained its sequence and coding DNA sequence (CDS) information from the Ensembl database [28]. We then identified cytosine substrates (potential risk substrates; PRSs) whose conversion to U(T) could introduce a premature stop codon and truncate the encoded protein. Each PRS was evaluated using our CBE-induced RNA off-target classifiers. We defined substrates predicted as high risk by our classifiers as effective PRSs. We then calculated the effective substrate density (ESD) (i.e., the number of effective PRSs divided by the total number of PRSs and the transcript’s amino acid length) and used the average ESD per tissue or organ as a representative measure of CBE-induced RNA off-target risk. When using the SNL model, we found that the brain, adipose tissue, ovaries, heart, and muscle exhibited significantly reduced RNA off-target risk compared with all transcripts showing tissue-specific expression, whereas the prostate, colon, and lungs showed significantly increased risk compared with all others (Figure 5). Similar trends were observed with the ACW/WCW motif-based classifiers and the STL model, consistently indicating relatively low risk in the brain and ovaries and relatively high risk in the colon and lungs. The colon, in particular, showed notably high risk.

When we extracted colon-specific transcripts with ESD values exceeding the 95% confidence interval of a t-distribution constructed from all tissue-specific transcripts, we found a total of 505 transcripts across all four classifiers (Appendix A). Of these, 202 transcripts (40%) were commonly identified by all four classifiers. The SNL model identified 365 colon-specific transcripts, including six unique transcripts not detected by the other classifiers. The ACW and WCW motif- and STL model-based classifiers identified 302 transcripts (24 unique), 380 transcripts (27 unique), and 393 transcripts (44 unique), respectively. We subjected these high-risk colon genes to Gene Ontology (GO) enrichment analyses to gain insight into their potential phenotypic impacts (Figure 6A, Appendix A). Seven GO terms, including GO:0095500, GO:1905145, GO:1905144, GO:0098926, GO:2000272, GO:0010469, and GO:0007267, were commonly enriched across all four classifiers (Figure 6B). These terms are associated with acetylcholine and signaling pathways. Specific high-risk genes, such as *LY6G6D* and *RGS10*, were related to some of these GO terms. Among the classifiers other than the ACW-based one, five common GO terms were enriched, including signaling-related terms such as GO:0038098, and a total of 14 GO terms were enriched in at least one non-ACW-based classifier. These results suggest that evaluating risk using only the ACW motif-based classifier, which suffers from both high false positive and false negative rates, may underestimate the true risk. Notably, we detected GO:0006978 (DNA damage response), whose status is obsolete, and the term was replaced by GO:0030330 on 2024-11-27, among the terms enriched in the SNL model. Recently, another group reported that in human hematopoietic stem/progenitor cells, the p53 response was lower for CBE (BE4max) than for Cas9 introduction [29]. A primary reason for this was likely the absence of double-strand breaks in CBE [30], whereas a minor possibility was that CBE-specific RNA off-target effects on transcripts associated with DNA damage response might affect the p53 response. Our results provide a foundation for designing more detailed follow-up analyses targeting *TP53* transcripts.

Notably, the STL and SNL models do not merely predict random substitution events but specifically target those derived from CBE-induced RNA off-target activity. The negative datasets used for the fine tuning contained sequences that exhibited substitutions regardless of CBE introduction. In actual cellular environments, spontaneously occurring background mutations coexist with CBE-induced events, making it ideal—within a base-editing risk assessment context—to employ models designed to isolate the impact of CBE-driven substitutions, thereby neither overestimating nor underestimating the associated risk.

This study showed that rAPOBEC1-based CBEs did not confine their off-target editing solely to ACW motifs, suggesting that RNA off-target dynamics may not rely on rigid protein–nucleotide interfaces. The broader WCW motif preferences implied that specific structural and biochemical features of A/T bases facilitate interactions with CBEs. This insight can guide future research on the structural dynamics of CBE-induced RNA off-targets. A previous study [21] described safer CBE variants with fewer RNA off-targets than BE4-rAPOBEC1. We suppose that the structural comparisons of CBE-to-A/T base affinity could be highly informative using the CBE variants. The structural insight from the assessment can contribute to the development of safer and more accurate CBEs.

One technical consideration warrants further research. All training data were derived from human RNA-seq datasets. While this approach is relevant for clinical applications of CBEs in humans, whether these findings can be generalized to other species remains to be validated. Despite the limitation, our study provides a quantifiable and standardized framework for assessing CBE-induced RNA off-target risks in human therapeutic contexts, forming a foundation for future research and broader CBE comparisons.

## 3. Materials and Methods

### 3.1. Implementation of PiCTURE

The PiCTURE pipeline comprises five steps: alignment, single-sample genotyping, joint genotyping, intersection, and mutation analysis (Figure 1).

PiCTURE processes RNA-seq data derived from base-editing samples. After alignment, a BAM file is generated and converted into a VCF file, from which substitution mutations and motifs are extracted. PiCTURE can handle various input types. For example, when using VCF files obtained from joint genotyping, PiCTURE can receive multiple RNA-seq datasets as input. Users can also analyze intersected VCF files in the “Mutation analysis” section. The hg38 reference genome, index files, interval lists, and human dbSNP data are retrieved from the GATK [31] resource bundle (https://gatk.broadinstitute.org/hc/en-us/articles/360035890811-Resource-bundle (accessed on 12 February 2025)).

PiCTURE can be run in both Docker [32] and Singularity (Apptainer) [33] containers, enabling operation on a variety of operating systems. Below, each process is described in detail.

#### 3.1.1. Alignment

Quality control and adapter trimming are carried out using Trim Galore v0.6.7 with Cutadapt [34]. The trimmed RNA reads are aligned to the hg38 human genome with STAR aligner v2.7.4a [35], using a two-pass alignment. The output is a sorted BAM file. All reads in the sorted BAM file are assigned to a new read group using AddOrReplaceReadGroups from Picard v2.3.0 [36]. Duplicate reads are marked with MarkDuplicatesSpark in GATK v4.3.0.0 [31]. Reads spanning splice junctions are split using SplitNCigarReads in GATK v4.3.0.0. Base quality recalibration is then carried out using baserecalibrator from GATK v4.3.0.0, generating a recalibration report, using the default setting for covariates and dbSNP build 138 [37] as a variant database for known sites.

#### 3.1.2. Single-Sample Genotyping

Genotypes are called using the HaplotypeCaller algorithm (-ERC GVCF) [38] in GATK v4.3.0.0 [31]. The variant calls from a single GVCF file are then aggregated into an efficient data structure via GenomicsDBImport in GATK v4.3.0.0. Subsequently, GenotypeGVCFs in GATK v4.3.0.0 is used for single-sample genotyping and to produce a single-sample VCF file.

#### 3.1.3. Joint Genotyping

Genotypes are called using the HaplotypeCaller algorithm (-ERC GVCF) in GATK v4.3.0.0. The variant calls from multiple GVCF files are aggregated into an efficient data structure via GenomicsDBImport in GATK v4.3.0.0. GenotypeGVCFs in GATK v4.3.0.0 is used for joint genotyping, resulting in a multi-sample VCF file containing only high-confidence variants.

#### 3.1.4. Intersection

An intersected VCF file can be obtained from the selected VCF files produced in Section 3.1.2 and Section 3.1.3. The selected VCF files are indexed with IndexFeatureFile in GATK v4.3.0.0. The indexed VCF files are compressed and merged using BCFtools v1.16 [39]. The intersected VCF is then extracted from the merged VCF file using SelectVariants (-select ’set == “Intersection”;’) in GATK v4.3.0.0].

#### 3.1.5. Mutation Analysis

Substitutions are extracted from the output VCF file using SelectVariants (--select-type-to-include SNP) from GATK v4.3.0.0. Next, substitutions are hard filtered with VariantFiltration from GATK v4.3.0.0. A substitution is filtered out if it meets any of the following criteria: QualByDepth  <  2, QUAL < 30, StrandOddsRatio > 3, FisherStrand  >  60, RMSMappingQuality  <  40, MappingQualityRankSumTest  < −12.5, and ReadPosRandSumTest  <  −8.

Qualified substitutions are then classified by substitution class using PySam v0.19.1 [40]. Motifs are generated by analyzing the substitution site plus 50 bp upstream and downstream flanking sequences with WebLogo v3.7.12 [41]. Variant Allele Frequencies (VAFs) are calculated with BCFtools v1.16 [39] (fill-tags FORMAT/VAF) as the fraction of reads with the alternate allele. The VCF files are split according to VAF thresholds (e.g., VAF ≥ 0.8 or VAF < 0.8), and motif generation is subsequently performed on these thresholded VCF files. The resulting substitution profiles are visualized using MultiQC v1.14 [42].

### 3.2. RNA-Seq Dataset

The RNA-seq data were acquired from the Sequence Read Archive. We obtained SRR8096262 in PRJNA498065 as BE3 transfection data [20]. For the BE4-rAPOBEC1 transfection data, we collected SRR11561273, SRR11561298, SRR11561324, SRR11561299, SRR11561297, and SRR11561323 from PRJNA595157 [21]. Additionally, we procured SRR11561326, SRR11561314, SRR11561289, SRR11561325, SRR11561303, and SRR11561278 as nCas9 transfection data for the negative controls. These RNA-seq data were obtained from cultured human HEK293T cells.

### 3.3. RNA Off-Target Analysis Using PiCTURE

All RNA-seq data were analyzed using the single-sample genotyping pipeline of PiCTURE. Nucleotide counts of substrates and their adjacent bases were aggregated from the statistical data output by the WebLogo function within PiCTURE. Statistical analyses with two-tailed Welch’s *t*-tests were conducted using custom Python scripts with SciPy [43]. Chromosome-scale visualization of variant allele frequency (VAF) data derived from substitution variants was achieved by processing the PiCTURE-generated VCF files with an R script using the “qqman” package [44]. All scripts were formatted for execution within a Snakemake workflow [45].

### 3.4. Preparation of the Fine-Tuning Dataset

For fine-tuning of the datasets, positive data were derived from RNA-seq samples transfected with BE3 or BE4-rAPOBEC1, whereas negative data were derived from RNA-seq samples transfected with nCas9. The construction of fine-tuning datasets for the STL and SNL models differed.

#### 3.4.1. Fine-Tuning Dataset for the STL Model

For training data, we used positive data (88,040 sequences) from SRR11561298 and negative data (21,790 sequences) from SRR11561314. From these samples, PiCTURE first identified each C-to-T substitution site and then extracted a 101-bp flanking region—comprising 50 bp upstream and 50 bp downstream—for each substituted site. Among the positive sequences, those overlapping with negative sequences were removed from the positive set and retained as negative sequences. These 9096 sequences were removed from the positive set. As a result, the training dataset comprised 78,944 positive and 21,790 negative sequences.

For the evaluation dataset, we used positive data (86,908 sequences) from SRR11561273 and negative data (24,228 sequences) from SRR11561326. Sequences were extracted in the same manner as for the training data, and any sequences overlapping between the positive and negative sets were removed from the positive set and retained in the negative set. Thus, 10,315 sequences were removed from the positive set. This yielded 76,593 positive and 24,228 negative sequences for evaluation.

For the test dataset, we used positive data from SRR11561324 (86,417 sequences) and negative data (23,476 sequences) from SRR11561289. Applying the above procedure resulted in a test set of 76,717 positive and 23,476 negative sequences. There were 9700 sequences removed from the positive set due to overlaps.

#### 3.4.2. Fine-Tuning Dataset for the SNL Model

First, we prepared a positive dataset by extracting C-to-T substitution sequences and their 20 bp upstream and 19 bp downstream flanking sequences from SRR8096262, SRR11561273, SRR11561297, SRR11561298, SRR11561299, SRR11561323, and SRR11561324 using PiCTURE. The sequences were merged into a single FASTA file (691,376 sequences). Similarly, we extracted negative data sequences from SRR11561278, SRR11561289, SRR11561303, SRR11561314, SRR11561325, and SRR11561326 and merged them into another FASTA file (139,187). We then removed duplicate sequences within positive_40bp.fa and negative_40bp.fa, using SeqKit v2.3.0 (rmdup) [46]. There were 155,474 and 70,807 sequences removed from the positive_40bp.fa and negative_40bp.fa, respectively.

Next, we used a custom Python script to identify sequences duplicated between the positive and negative sets. There were 26,461 duplicate sequences removed from the positive data and retained in the negative data. Using the train_test_split function in scikit-learn v1.3.0 [47], we split the positive and negative data at a ratio of 7:1.5:1.5 to create training, evaluation, and test datasets, respectively. We further balanced the datasets so that each subset contained equal numbers of positive and negative sequences. The final training, evaluation, and test datasets contained 47,866 positive and 47,866 negative sequences, 10,257 positive and 10,257 negative sequences, and 10,257 positive and 10,257 negative sequences, respectively.

### 3.5. Fine-Tuning a Pre-Trained Language Model for Predicting CBE-Based RNA Off-Targets

We fine-tuned the DNABERT-2 model (zhihan1996/DNABERT-2-117M from the Hugging Face repository) [22] to create a classifier capable of detecting C-to-U substitutions induced by CBEs on RNA, yielding two models: the STL model and the SNL model.

#### 3.5.1. Fine-Tuning Parameters for the STL Model

The total number of original sequences from the PiCTURE output was 330,859 sequences. The sequence dataset was split by RNA-seq data and filtered as described in Section 3.4.1. We used 78,944, 76,593, and 76,717 sequences for training, evaluation, and test as positive datasets, respectively. We also used 21,790, 24,228, and 23,476 sequences for training, evaluation, and test as negative datasets, respectively.

We fine-tuned zhihan1996/DNABERT-2-117M, with the following parameters: model_max_length = 10, per_device_train_batch_size = 10, per_device_eval_batch_size = 16, gradient_accumulation_steps = 1, learning_rate = 3 × 10^−5^, fp16 = TRUE, num_train_epochs = 10, save_steps = 200, evaluation_strategy = steps, eval_steps = 200, warmup_steps = 55, and logging_steps = 100.

#### 3.5.2. Fine-Tuning Parameters for the SNL Model

The total number of original sequences from the PiCTURE output was 830,563 sequences. The sequence dataset was split by positive and negative labels and filtered as described in Section 3.4.2. We used 47,866, 10,257, and 10,257 sequences for training, evaluation, and test as positive datasets, respectively. We also used 47,866, 10,257, and 10,257 sequences for training, evaluation, and test as negative datasets, respectively.

We fine-tuned zhihan1996/DNABERT-2-117M, with the following parameters: model_max_length = 10, per_device_train_batch_size = 8, per_device_eval_batch_size = 16, gradient_accumulation_steps = 8, learning_rate = 2e−5, fp16 = TRUE, num_train_epochs = 8, save_steps = 200, evaluation_strategy = steps, eval_steps = 200, warmup_steps = 50, and logging_steps = 100.

Accuracy, precision, recall, and F1 scores for the fine-tuned models were obtained from the output of the DNABERT-2 train.py script. Additionally, we evaluated accuracy, precision, recall, and F1 scores for ACW, WCW, and NNN motif-based models using the test data via a custom script that leveraged scikit-learn v1.4.2.

The STL model (https://huggingface.co/KazukiNakamae/STLmodel (accessed on 12 February 2025)) and SNL model (https://huggingface.co/KazukiNakamae/SNLmodel (accessed on 12 February 2025)) were deposited on the Hugging Face platform [48].

### 3.6. Comparing Classifier Performance for Canonical and Non-Canonical RNA Off-Targets

Using custom Python scripts, we collected 20,514 sequences (40 bp each) from the SNL model test dataset and divided them into positive and negative sets. Within each set, the sequences were further divided into those with and those without the ACW motif, generating a canonical RNA off-target dataset with 4987 positive and 1498 negative sequences, and a non-canonical RNA off-target dataset with 5270 positive and 8759 negative sequences. Both datasets were evaluated using the ACW and WCW motif classifiers and the STL and SNL models.

After performing the predictions, we randomly selected 1000 positive and 1000 negative sequences from each dataset, fixing the random seed for reproducibility. Using scikit-learn v1.5.1, we calculated accuracy, precision, recall, and F1 scores. We then aggregated these results and conducted two-tailed Welch’s *t*-tests using SciPy v1.10.1.

### 3.7. Tissue-Specific Risk Assessment Using PROTECTiO

We applied PROTECTiO to evaluate how CBE-induced RNA off-target events might affect genes with tissue- or organ-specific elevated expression.

First, we obtained tissue-specific expression patterns that were pre-calculated from Illumina bodyMap2 transcriptome RNAseq data (PRJEB2445) in RefEx (https://refex.dbcls.jp (accessed on 12 February 2025)) [27], a curated gene expression reference dataset. This dataset is publicly available at https://dx.doi.org/10.6084/m9.figshare.4028709 (accessed on 12 February 2025). The expression patterns were organized by NCBI RefSeq IDs [49], with tissue specificity indicated by binary values (1: tissue-specific high expression, −1: tissue-specific low expression). We then retrieved the corresponding Ensembl transcript IDs for each NCBI RefSeq [49] ID using the TogoID [50] API. Using the Ensembl [28] REST API, we obtained CDS coordinates and genomic sequence information for each transcript and reconstructed the CDS sequences. From these CDS sequences, we deduced amino acid sequences and identified PRSs, i.e., cytosine residues whose C-to-U substitution could introduce a premature stop codon.

For each PRS identified, we retrieved the 20 bp upstream and 19 bp downstream flanking sequences from the genomic coordinates and listed them using Ensembl [28] transcript ID. We then evaluated these sequences with the ACW and WCW motif classifiers and STL and SNL model classifiers, classifying any PRS predicted as positive by a classifier as an effective PRS. To compute the ESD, we summed the numbers of effective PRSs per transcript and normalized the sums to the total number of PRSs in that transcript and to the length (in amino acids) of the encoded protein.

We aggregated ESD values per tissue and conducted two-tailed t-tests using SciPy v1.10.1 and statsmodels v0.14.1 [51]. Transcripts whose ESDs exceeded the 95% confidence interval of a t-distribution constructed from all tissue-specific transcripts were defined as high-risk genes. The transcript list was exported as a CSV file.

### 3.8. Enrichment Analysis of High-Risk Genes

We inputted the list of high-risk genes generated by PROTECTiO into ShinyGO 0.81 (http://bioinformatics.sdstate.edu/go/ (accessed on 12 February 2025)) [52] for enrichment analysis. We focused only on GO [53] terms related to Biological Processes, setting the FDR cutoff at 0.05 and the pathway size between 2 and 2000. For ontology analysis, we used the “simona” v1.2.0 package [54] to assess computing term similarity using the “Sim_WP_1994” method [55]. Word clouds were generated using the “simplifyEnrichment” package [56], and an UpSet plot was created using the “Complexupset” package [57].

## 4. Conclusions

We developed PiCTURE, a standardized pipeline for analyzing CBE-induced RNA off-targets, and successfully detected both canonical (ACW motif-containing) and non-canonical RNA off-targets. By integrating language model-based machine learning, we accurately predicted non-canonical RNA off-target events, overcoming the limitations of motif-only classifiers. When applying our models to different human tissues, the results suggested a relatively low RNA off-target risk in the brain and ovaries and a relatively high risk in the colon and lungs. This tissue-specific risk information can serve as a valuable resource for predicting potential side effects of CBEs and guiding safety evaluations in clinical applications.

## Figures and Tables

**Figure 1 ijms-26-01723-f001:**
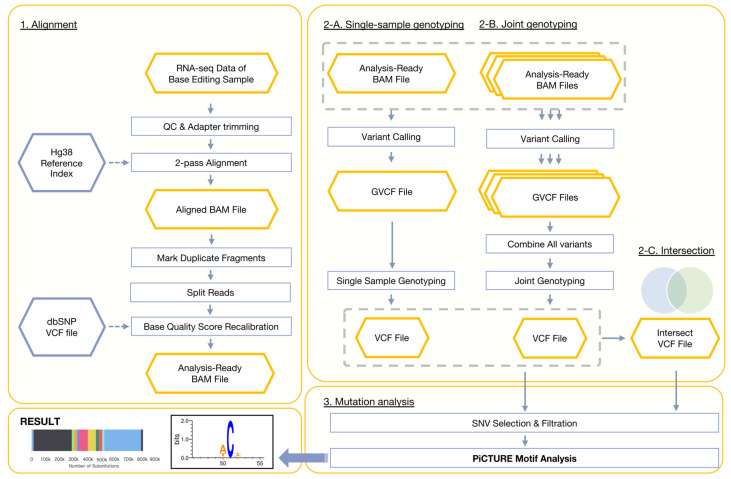
Overview of the PiCTURE pipeline for RNA off-target analysis. The PiCTURE workflow for detecting and characterizing CBE-induced RNA off-target events is shown. (1) Input RNA-seq data from base editing samples undergo quality control, adapter trimming, and two-pass alignment to the hg38 reference genome. After alignment, duplicate fragments are marked, reads spanning splice junctions are split, and base quality scores are recalibrated using the dbSNP database, resulting in an analysis-ready BAM file. (2A) In single-sample genotyping, variants are called from the analysis-ready BAM file, producing a GVCF file and, after genotyping, a single-sample VCF file. (2B) For joint genotyping, variants from multiple GVCFs are combined to generate a multi-sample VCF, and (2C) intersected VCF files can be produced by selecting variants common to multiple conditions. (3) During mutation analysis, SNVs are filtered, and motif analyses are conducted using the integrated tools in PiCTURE, yielding final outputs such as substitution profiles and sequence motif representations. Yellow hexagon box represents files.

**Figure 2 ijms-26-01723-f002:**
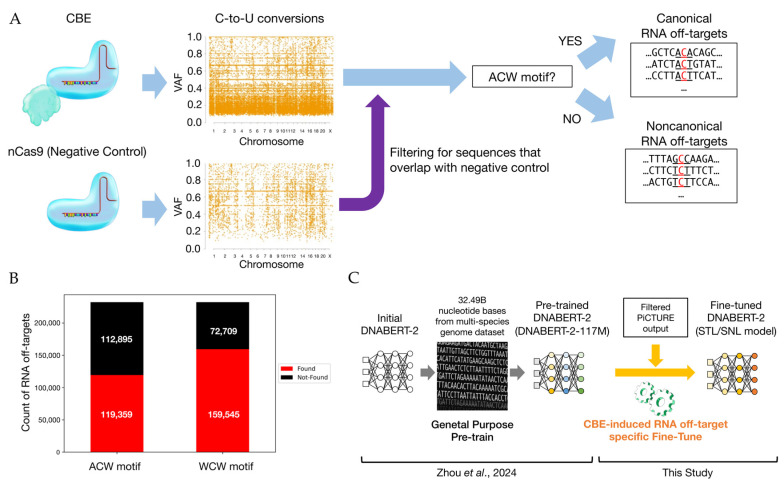
Conceptual workflow for CBE-based RNA off-target motif identification and the fine-tuning of a language model. The process for identifying and characterizing RNA off-target events caused by CBEs and subsequently integrating this information into a machine learning framework is shown. (**A**) Starting from RNA-seq data, PiCTURE detects numerous C-to-U conversions, which can be classified into canonical and non-canonical RNA off-targets based on the ACW motif. A purple arrow represents filtering for sequences that overlap with negative control. (**B**) The numbers of RNA off-targets associated with ACW and WCW motifs are quantified and compared. “Found” refers to sequences in which the specified motif (e.g., the ACW motif) is present at the identified C-to-U substitution site. “Not Found” indicates that the motif is absent at that location. (**C**) The sequence datasets from PiCTURE are then used to fine-tune the DNABERT-2 model for predicting CBE-induced RNA off-targets. The DNABERT-2 model used is pre-trained on a large-scale genomic corpus (32.49B nucleotide bases from a multi-species genome dataset) created by another group (Zhou et al., 2024 [22]). An orenge arrow represents CBE-induced RNA off-target specific Fine-Tune in this study.

**Figure 3 ijms-26-01723-f003:**
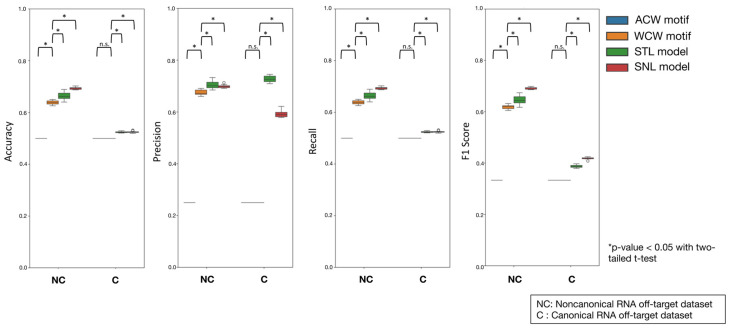
Performance comparison of motif- and language model-based classifiers for canonical and non-canonical RNA off-target detection. Box plots show the accuracy, precision, recall, and F1 score distributions for the ACW and WCW motif classifiers and STL and SNL models. Each classifier’s performance was evaluated using datasets containing canonical (ACW motif) and non-canonical (no ACW motif) RNA off-targets. Asterisks indicate *p* < 0.05 with two-tailed Welch’s *t*-test and “n.s.” denotes no significant difference.

**Figure 4 ijms-26-01723-f004:**
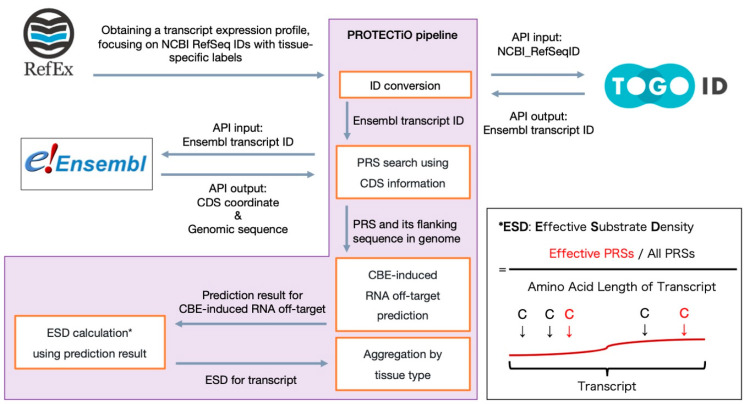
Overview of the PROTECTiO pipeline for tissue-specific RNA off-target risk assessment. The workflow employed by PROTECTiO to evaluate tissue-specific CBE-induced RNA off-target risks is shown. Tissue-specific gene expression data were retrieved from RefEx and mapped to corresponding Ensembl transcript IDs via TogoID in the ID conversion process. Using the Ensembl REST API, CDSs and associated genomic coordinates were obtained, enabling the extraction of CDS sequences. Each candidate cytosine (C), named PRS, that could form a premature stop codon when deaminated was identified using the CDS and genomic sequences, and PRS flanking sequences were evaluated using various classifiers (motif- or language model-based) for CBE-induced RNA off-target prediction. The results were then processed to calculate the ESD for the transcript. The formula used to calculate the ESD, which reflects the relative abundance of effective PRSs in each transcript, is provided in the right box (indicated by an asterisk). Red arrows indicate effective PRSs. Finally, all effective substrates corresponding to transcripts significantly upregulated in at least one tissue or organ were grouped by tissue type and two-tailed Welch’s *t*-tests were then performed to estimate organ- and tissue-specific RNA off-target risks.

**Figure 5 ijms-26-01723-f005:**
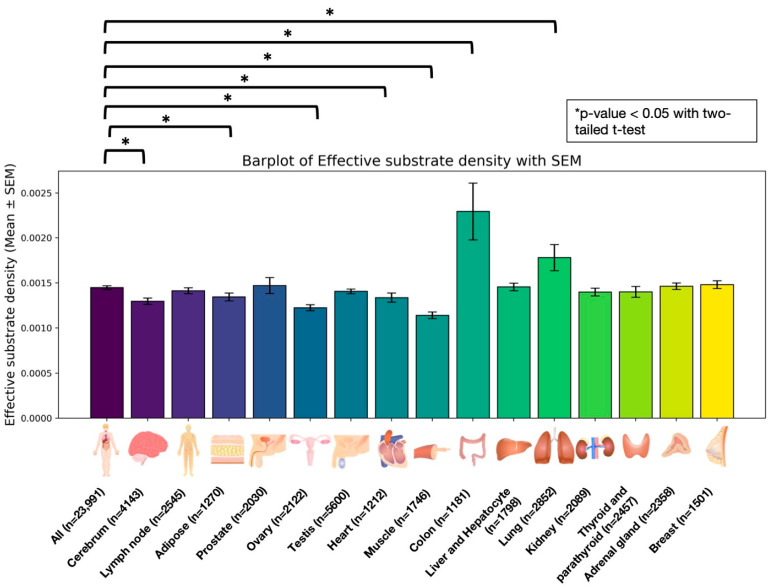
Tissue-specific ESD illustrating RNA off-target risk. The bar plot displays the average ESD of transcripts with standard error of the mean across various human tissues. The “All” column lists transcripts that are tissue-specifically upregulated or downregulated in at least one of the tissues and organs. The other columns list tissue-specific active transcripts, which are upregulated transcripts in the tissues or organs. Lower ESD values indicate reduced predicted RNA off-target risk, whereas higher values suggest an elevated risk. “n” represents the number of transcripts. Asterisks indicate *p* < 0.05 with two-tailed Welch’s *t*-tests. The illustrations are from TogoTV (© 2016 DBCLS TogoTV, CC-BY-4.0).

**Figure 6 ijms-26-01723-f006:**
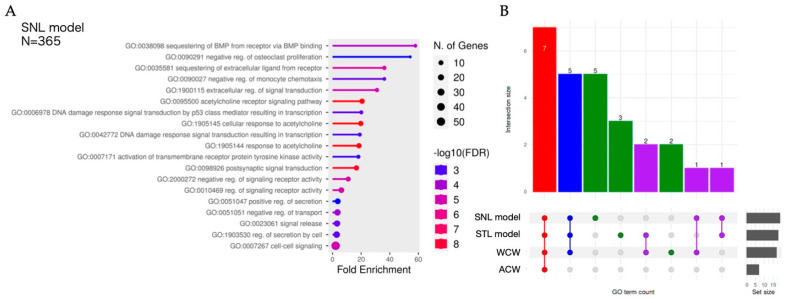
Enrichment analysis of GO terms associated with high-risk genes identified by multiple classifiers. Summary of enrichment analysis. (**A**) The figure presents Gene Ontology (GO) enrichment analyses for high-risk, colon-specific transcripts (genes) identified by the SNL model. The plot shows enriched GO Biological Process terms along with their fold enrichment and significance (-log10(FDR)). Each bar corresponds to a GO term, color-coded according to its –log10(FDR) value, while the dot size indicates the number of associated genes. The N represents the number of high-risk colon-specific transcripts. (**B**) The upset plot of GO terms commonly enriched in different classifier sets. Colored bars represent the count of GO terms found in one or more classifier sets (SNL model, STL model, WCW motif, and ACW motif). Dots with connecting lines indicate classifier combinations.

**Table 1 ijms-26-01723-t001:** Performance comparison of motif-based classifiers and the STL model for RNA off-target prediction.

	Accuracy	Precision	Recall	F1 Score
STL model	0.803	0.729	0.672	0.690
ACW motif classifier	0.596	0.636	0.686	0.580
WCW motif classifier	0.703	0.663	0.721	0.662
NNN motif classifier	0.766	0.383	0.5	0.434

**Table 2 ijms-26-01723-t002:** Performance comparison of motif-based classifiers and the SNL model for RNA off-target prediction.

	Accuracy	Precision	Recall	F1 Score
SNL model	0.726	0.729	0.726	0.725
ACW motif classifier	0.670	0.697	0.670	0.659
WCW motif classifier	0.717	0.717	0.717	0.716
NNN motif classifier	0.5	0.25	0.5	0.333

## Data Availability

Publicly available datasets were analyzed in this study. The script for PiCTURE and the associated scripts are available here: https://github.com/KazukiNakamae/PiCTURE (accessed on 21 December 2024). The STL model is available here: https://huggingface.co/KazukiNakamae/STLmodel (accessed on 21 December 2024). The SNL model is available here: https://huggingface.co/KazukiNakamae/SNLmodel (accessed on 21 December 2024). The script for PROTECTiO and the associated scripts are available here: https://github.com/KazukiNakamae/PROTECTiO (accessed on 21 December 2024). The transcript-level evaluation data generated by PROTECTiO are publicly available here: https://doi.org/10.6084/m9.figshare.28053845.v1 (accessed on 21 December 2024).

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
