# Peer review of "Risk Prediction of RNA Off-Targets of CRISPR Base Editors in Tissue-Specific Transcriptomes Using Language Models"

_ijms, 2025, doi:10.3390/ijms26041723_

Round 1
Reviewer 1 Report
Comments and Suggestions for Authors
The authors analyzed samples treated with BE3 using the Picture pipeline. They found that the sequence motifs that influence specificity are WCW with a preponderance of ACW, which is in agreement with other previous studies. Using this result as validation of Picture, other samples treated with BE4-rAPOBEC1 were analyzed, finding similar results for the conversion sites.
With the detected and undetected sequences, a neural network (DNABERT-2) was trained using two strategies: 101 bp sequences (STL) and 40 bp sequences (SNL). The first model leads to a better performance and both models outperform previous results.
I think the manuscript deserves to be published in IJMS, I only have some minor recommendations
minor points
1.- Although it is clearly written, I do not understand the reason why they use a negative dataset derived from the nCAS9 samples and the positive datasets. Why do they not use the datasets corresponding to the same sample to train the language models? Since the efficiency of neural network models strongly depends on the training set, I think the authors should try other strategies.
2.- What is the meaning of "found" an "not found" in figure 2B?
3.- There are typos in line 442
Author Response
Comment-1:
Although it is clearly written, I do not understand the reason why they use a negative dataset derived from the nCAS9 samples and the positive datasets. Why do they not use the datasets corresponding to the same sample to train the language models? Since the efficiency of neural network models strongly depends on the training set, I think the authors should try other strategies.
Response-1:
We greatly appreciate this insight. We used substitution data from nCas9-treated RNA-seq samples as the negative dataset to account for spontaneous mutations that are independent of CBEs. Because nCas9 lacks both the functional nuclease domain responsible for double-strand breaks and the deaminase domain, RNA-seq data from nCas9-transfected samples primarily reflect naturally occurring background substitutions rather than editing-specific events. If we were to use the same CBE-treated RNA-seq data in both the positive and negative sets, we would be unable to distinguish CBE-induced deaminations from other deamination processes. As mentioned in lines 322–329, we focused on developing a model that specifically learns to identify CBE-induced substitutions rather than all possible substitution events, thereby preventing the overestimation of CBE-related risks. We have included the following additional description at lines 153–158:
“In this development, we focused on a model that specifically learns to identify ‘CBE-induced substitutions’ rather than all possible substitution sites. We used nCas9-treated RNA-seq samples as the negative dataset to capture background mutations that are independent of CBEs, ensuring our classifier targets genuinely CBE-driven effects and avoids under- or overestimating CBE-associated risk.”
We agree that exploring alternative negative datasets and strategies would be valuable for future work, especially for broader studies of various RNA substitutions beyond CBEs.
Comment-2:
What is the meaning of "found" an "not found" in figure 2B?
Response-2:
Thank you for pointing out the potential ambiguity. In Figure 2B, “Found” refers to sequences in which the specified motif (e.g., the ACW motif) is present at the identified C-to-U substitution site. “Not Found” indicates that the motif is absent at that location. We have clarified this wording in the revised figure legend to avoid confusion.
Comment-3:
There are typos in line 442
Response-3:
We have corrected these typos in the revised manuscript. Specifically, the sentence at line 458 (line 442 in the previous version) now reads “sequences” instead of “sequneces.”
Reviewer 2 Report
Comments and Suggestions for Authors
The manuscript “Risk Prediction of RNA Off-Targets of CRISPR Base Editors in 2 Tissue-Specific Transcriptomes Using Language Models” by Kazuki Namkamae et al. describes the use of language models to determine base editing sites specifically cytosine editors using single strand break CRISPR editors (nCas9 as a negative control and BE4-rAPOBEC1 for motif substitutions). In particular, the focus was on canonical editing, so-called ACW motif sites where W=A or T/U and WCW non-canonical sites where editing also takes place. Their pipeline PiCTURE (integrated with DNABERT-2) was developed to make these determinations. The authors also describe the integration of PiCTURE outputs with the PROTECTiO pipeline to estimate the RNA off-target editing effects for transcripts showing tissue specific expression. Interestingly, their results show relatively low off-target effects for the brain and ovaries while exhibiting high off-target effects for the colon and lungs. These targeting effects can introduce premature stop codons or correct pathogenic SNPs. Their pipelines when compared to classifiers that focused on ACW, WCW and NNN motifs mostly outperformed these focused classifiers in terms of accuracy, precision and F1 (combined recall and precision). They further improved their results by balancing their data set for positive and negative sequences producing a classifier that outperformed the other focused classifiers in all these measurements.
The paper is well written and pays significant attention to the construction of training and testing data sets and analysis. The results appear to be helpful for determining the effect of the nCas9 editors on specific genes and their potential ability to be used as therapeutic agents. The only suggestion for possible inclusion in the manuscript is can the authors provide a structural interpretation for the somewhat promiscuous editing that occurs due to the variations in motif, and if that can be described, are there any modifications that can be suggested to further improve the accuracy of the CRISPR editors.
Author Response
Comments 1
The paper is well written and pays significant attention to the construction of training and testing data sets and analysis. The results appear to be helpful for determining the effect of the nCas9 editors on specific genes and their potential ability to be used as therapeutic agents. The only suggestion for possible inclusion in the manuscript is can the authors provide a structural interpretation for the somewhat promiscuous editing that occurs due to the variations in motif, and if that can be described, are there any modifications that can be suggested to further improve the accuracy of the CRISPR editors.
Responses 1
We appreciate this thoughtful suggestion. Although our study focuses on transcriptome-wide analyses rather than structural biology, we agree that structural considerations are crucial for understanding motif variations and editing promiscuity. Our findings suggest that rAPOBEC1-based CBEs do not strictly target ACW motifs on RNA, implying that CBE-induced RNA off-target dynamics may not be dictated solely by rigid protein–nucleotide interactions. Instead, the broader WCW motif preferences we observed indicate that the structural and biochemical properties of A/T bases may enhance interactions with CBEs. A previous study (Yu et al., Nature Communications, 2020) reported several safer CBE variants with fewer RNA off-targets than BE4-rAPOBEC1. Using the CBE variants, It is worth comparing structural features that influence CBE-to-A/T affinity. We have added a brief note on lines 331–339:
“This study showed that rAPOBEC1-based CBEs did not confine their off-target editing solely to ACW motifs, suggesting that RNA off-target dynamics may not rely on rigid protein–nucleotide interfaces. The broader WCW motif preferences implied that specific structural and biochemical features of A/T bases facilitate interactions with CBEs. This insight can guide future research on the structural dynamics of CBE-induced RNA off-targets. A previous study [21] had described safer CBE variants with fewer RNA off-targets than BE4-rAPOBEC1. We suppose that the structural comparisons of CBE-to-A/T base affinity could be highly informative using the CBE variants. The structural insight from the assessment can contribute to the development of safer and more accurate CBEs”